



# Age-depth distribution in western Dronning Maud Land, East Antarctica, from three decades of radar surveys

Steven Franke[1,2], Daniel Steinhage[2], Veit Helm[2], Alexandra M. Zuhr[1], Julien A. Bodart[3], Olaf Eisen[2,4], and Paul Bons[1]

[1]Department of Geosciences, Tübingen University, Tübingen, Germany
[2]Alfred-Wegener-Institut Helmholtz-Zentrum für Polar- und Meeresforschung , Bremerhaven, Germany
[3]Climate and Environmental Physics, Physics Institute and Oeschger Centre for Climate Change Research, University of Bern, Bern, Switzerland
[4]Department of Geosciences, University of Bremen, Bremen, Germany

**Correspondence:** Steven Franke (steven.franke@uni-tuebingen.de)

**Abstract.** Radio-echo sounding provides the opportunity to study the internal architecture of ice sheets through imaging stratified englacial reflections, known as internal reflection horizons (IRHs). They represent consistent time horizons formed at the former ice-sheet surface and buried over time, thus reflecting the ice sheet's age–depth architecture. Their analysis allows crucial insights into past and present boundary conditions, e.g. accumulation rates or basal melting, as well as physical properties

and ice dynamics. This study presents a comprehensive data set and insight into the age–depth distribution in western Dronning Maud Land (DML), East Antarctica, spanning the Holocene to the Last Glacial Period (4.8 – 91.0 ka). Using data from various radar systems deployed by the Alfred Wegener Institute between 1996 and 2023, we traced and dated nine IRHs over an area of 450 000 km². A precise age could be assigned to the IRHs by two-way travel time to depth conversion and employing radar forward modelling based on conductivity peaks of the EPICA DML ice core. Six IRHs correlate with past volcanic eruptions.

Our findings suggest that most IRHs correspond to IRHs of similar age in other regions of East and West Antarctica, thus likely originating from the same physical reflectors at depth, although they could not be physically connected. This work enhances understanding of the englacial architecture and relationships with snow accumulation and ice-dynamic processes of this sector of the Antarctic ice sheet and provides fundamental data for numerical ice flow models and paleoclimatic studies.

## 1 Introduction

Observing and modelling the Antarctic ice sheet (AIS) is crucial for understanding both past and present climate dynamics and improving projections of future sea-level changes (IPCC, 2023). Deducing its past evolution helps us to comprehend the processes driving ice flow, accumulation, and melting. As one unique observable property, the ice sheet's internal stratigraphy represents a valuable record of past kinematics and dynamics (Siegert et al., 2004; Bons et al., 2016; Leysinger Vieli et al., 2018; Jansen et al., 2024), offering insights into atmosphere–ice–ocean interactions (Drews et al., 2020; Višnjević et al., 2022)

and how the ice-sheet system responded to natural climatic variations over a broad range of timescales ranging from hundreds to hundred thousands of years (e.g., Leysinger Vieli et al., 2004; Sutter et al., 2021; Bodart et al., 2023).





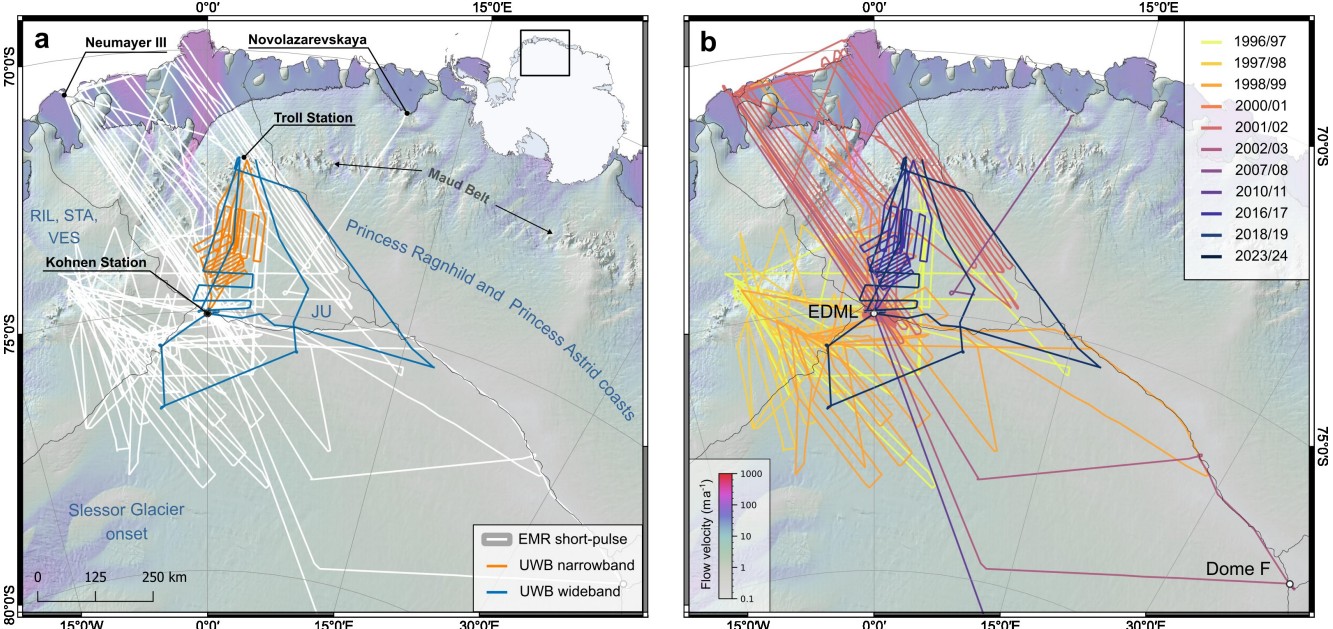

**Figure 1.** Radar profiles used for IRH tracing. (a) Radar profiles are sorted by system: EMR short-pulse (white lines), MCoRDS narrowband (orange lines), and MCoRDS wideband (blue lines). (b) Radar data sorted by acquisition season. Although some profiles extend down to the ice shelf, the IRH tracing region is restricted upstream of the Maud Belt. The drainage basins are marked by thin black lines and are from Zwally et al. (2012).

Radar data provide the opportunity to study the internal architecture of ice sheets (i.e., its stratigraphy) via continuous stratified englacial reflections. These internal reflection horizons (IRHs) occur when radar waves are reflected at boundary layers where the dielectric properties of the ice change (Glen and Paren, 1975; Robin et al., 1977). Dielectric contrasts in ice

sheets are primarily caused by variations in density (Robin et al., 1969), electrical conductivity (Paren and Robin, 1975), and changes in the preferred orientation of ice crystals (Harrison, 1973; Fujita et al., 1999). IRHs that are located sufficiently deep below the ice surface, typically within a few hundred meters in Antarctica, are primarily linked to conductivity contrasts and can sometimes be detected across the ice sheet (Millar, 1981; MacGregor et al., 2015; Winter et al., 2019). Strong conductivity contrasts are often linked to deposits of acidic materials from major volcanic eruptions (Hammer, 1980; Millar, 1981). Thus,

IRHs represent boundary layers in the ice that indicate time horizons of the same age of snow deposits.

Information about the internal stratigraphy and age–depth architecture of the Antarctic ice sheet is crucial as it serves as a climatic and ice-dynamic record. For instance, it provides insights into past accumulation rates and basal melting, informing us about past surface and basal mass balance (Leysinger Vieli et al., 2004; Eisen et al., 2005; Leysinger Vieli et al., 2011; Cavitte et al., 2018; Bodart et al., 2023; Koch et al., 2023). Since many IRHs can be dated with ice cores, they can also serve

as calibration points for ice sheet models (Sutter et al., 2021; Višnjević et al., 2022) and to model basal conditions (Leysinger Vieli et al., 2018; Fudge et al., 2023; Chung et al., 2023; Wang et al., 2023). Additionally, the geometry, depth and continuity



**Table 1.** AWI radar system specifications.

| Radar system | Developper | Frequency | Transmit signal | Range resolution | Seasons |
|---|---|---|---|---|---|
| EMR | TU HH[a] | 150 MHz | 60 ns burst | ∼5 m | 1996/97, 1997/98, 1998/99<br>2000/01, 2001/02, 2002/03<br>2007/08, 2010/11, 2016/17 |
| AWI MCoRDS 5 | CReSIS[b] | 180 – 210 MHz<br>(narrowband) | 1, 3 & 10 μs chirp | 4.3 m | 2018/19 |
| MCoRDS MCoRDS 5 | CReSIS[b] | 150 – 520 MHz<br>(wideband) | 1, 3 & 10 μs chirp | 0.35 m | 2023/24 |

[a] Technical University Hamburg-Haburg

[b] Center for Remote Sensing and Integrated Systems

of IRHs contain information on the cumulative deformation due to ice flow, making them suitable passive markers for ice-dynamic activity and its changes over time, e.g., deciphering folding processes and ice stream activity (Siegert et al., 2004; Bons et al., 2016; Franke et al., 2022a; Jansen et al., 2024), and ice-dynamic processes causing layer discontinuity (Panton and
Karlsson, 2015; Sanderson et al., 2023).

The generation and compilation of IRHs using radar data in Antarctica and Greenland remains largely a manual task. While automated methods can successfully trace IRHs to a certain extent (Moqadam and Eisen, 2024), these algorithms often fail to consistently follow the same reflection across multiple intersecting radar profiles and different radar systems, where reflections are depicted differently. Nevertheless, IRHs have been traced and dated across large parts of East Antarctica (Winter et al., 2019;
Cavitte et al., 2016, 2021; Wang et al., 2023), West Antarctica (Siegert et al., 2005; Jacobel and Welch, 2005; Ashmore et al., 2020; Bodart et al., 2021; Beem et al., 2021; Muldoon et al., 2018), and Greenland (Karlsson et al., 2013; MacGregor et al., 2015; Franke et al., 2023a). For Antarctica, in particular, these studies have been motivated by the AntArchitecture initiative, a Scientific Committee of Antarctic Research's Action Group which aims to build a 3-D age–depth model of Antarctica from IRHs.

In this study, we present a comprehensive and detailed insight into the dated radiostratigraphy of western Dronning Maud Land (DML; East Antarctica), spanning from the Holocene to the Last Glacial Period, using radar data collected over the past three decades by various recording systems of the Alfred Wegener Institute, Helmholtz Centre for Polar and Marine Research (AWI; Figure 1). Through radar forward modeling based on conductivity data from the EPICA DML (EDML) ice core, we can accurately date IRHs and, in some cases, link them to deposits from past volcanic eruptions. We also discuss the extent
to which our dated IRHs can be correlated with those from other Antarctic-wide studies and demonstrate the potential for extrapolating some of these IRHs to much larger areas where an AWI radar system with coarser vertical resolution was used.





## 2 Data and Methods

### 2.1 Radar data

In this study, we utilize three types of radar products that provide sufficiently high range resolution (0.35 – 5 m) and penetration
depth. These radar products were obtained during campaigns conducted by AWI, between 1996 and 2023 with two different
radar systems.

#### 2.1.1 AWI EMR short-pulse data

The largest pool of radar data is provided by AWI's EMR (Electromagnetic Radar) system, which has been operational in
Antarctica and Greenland since 1994 (Nixdorf et al., 1999). For IRH tracing, we utilize EMR radar data from nine Antarctic
seasons spanning from 1996/97 to 2016/17 (Figure 1 and Table 1). The EMR system consists of two dipole antennas mounted
underneath the wings of AWI's Polar Aircrafts (Alfred-Wegener-Institut Helmholtz-Zentrum für Polar- und Meeresforschung,
2016). The transmission signal is a burst with pulse lengths of 60 or 600 ns at a frequency of 150 MHz, with a pulse repetition
frequency (PRF) of 20 kHz. For this study, we exclusively utilize EMR 60 ns pulse data (short-pulse data) due to their higher
range resolution to detect IRHs compared to the 600 ns pulse data (see Wang et al., 2023). The short-pulse data have a range
resolution of approximately 5 m and a vertical sample interval of 13.33 ns. EMR radar data processing comprises filtering and
along-track stacking with a factor of 7 or 10, which results in a mean trace spacing of ∼ 35 and ∼ 50 m, respectively (Steinhage
et al., 2001; Steinhage, 2001; Wang et al., 2023).

EMR survey lines cover the entire western DML (Figure 1). Surveys between 1997 and 1999 focused on the pre-site survey
of the EDML ice core site (Steinhage et al., 2001; Steinhage, 2001), while surveys between 2001 and 2011 were used to
investigate the internal structure (Eisen et al., 2006, 2007; Drews et al., 2009; Steinhage et al., 2013) as well as ice thickness
for solid earth geophysical studies in DML (Riedel et al., 2012; Eisermann et al., 2020). The survey in 2016/17 extensively
surveyed ice thickness at the Dome Fuji ice core site (Karlsson et al., 2018; Wang et al., 2023).

#### 2.1.2 AWI multi-channel chirp data

In addition, we use two data sets acquired with AWI's multi-channel airborne chirp radar system. The system is an improved
version of the Multichannel Coherent Radar Depth Sounder (MCoRDS, version 5; here refered to as AWI MCoRDS system),
which was developed at the Center for Remote Sensing of Integrated Systems (CReSIS) at the University of Kansas (Rodriguez-
Morales et al., 2013; Hale et al., 2016) and has been in operation since 2016 (Kjær et al., 2018). The radar configuration for
both campaigns in 2018/19 and 2023/24 consisted of an eight-element antenna array mounted under AWI's Polar 5 or Polar 6
Basler BT-67 aircraft's fuselage, which serves as transmit and receive antenna array. The transmission signal is composed
of several staged modulated chirp signals, which provide high resolution at different depths (Franke et al., 2022b). Standard
processing techniques were performed with the OPR Toolbox (Open Polar Radar Toolbox; formerly termed CReSIS Toolbox;
Open Polar Radar, 2023). The main steps include motion compensation, pulse compression, synthetic aperture radar (SAR)



focusing (in the fk domain) and array processing. Airplane location and orientation is from Global Positioning System (GPS) precise point positioning (PPP) post-processed with a final estimated accuracy (commercial software package Waypoint 8.4) of better than 3 cm for latitude and longitude and better than 10 cm for altitude.

The two AWI MCoRDS campaigns were flown in two different acquisition settings in the 2018/19 and 2023/24 seasons, respectively. In the 2018/19 season, data were acquired with a frequency range between $180 - 210$ MHz (narrowband), which corresponds to a range resolution of 4.3 m. The acquisition settings were optimized to sound deep englacial reflections as well as the bed topography at the onset of Jutulstraumen Glacier (Franke et al., 2021, 2024). MCoRDS data in the 2023/24 season was acquired with a frequency range between $150 - 520$ MHz (wideband), with a range resolution of $\sim 35$ cm. The acquisition settings were chosen to map near-surface IRHs at high spatial resolution (Koch et al., 2023) in the upstream part of the Jutulstraumen catchment as well as the central plateau South and West of the EDML ice core.

## 2.2 Internal reflection horizon tracing

For tracing IRHs, we utilize the commercial software Emerson Paradigm. To enhance the visibility of isochrones at greater depths, we applied an Automatic Gain Control (AGC) algorithm to the radargrams. IRHs were traced with the following rationale: (1) Existing IRHs, such as the 38 ka and 74 ka isochrones from Winter et al. (2019), were extended, (2) the shallowest and deepest clearly identifiable reflector in the EMR short-pulse data was traced, (3) additional isochrones at regular intervals between those defined in (1) and (2), which are clearly discernible in all radar products.

All isochrones were manually traced with the assistance of a semi-automatic picker, which searches for the maximum within a window width of 5 ns along two picks. An additional gain function was applied to the radar data (also known internally as AGG products) to enhance reflections at all depths. Based on the intersection points of the radar profiles, IRHs were transferred between the different radar products. In cases where the assignment at the intersection point was not clear or the radar reflector was no longer traceable within a profile, no picks were created. We clarify that we use the existing IRHs from Winter et al. (2019) in DML for our representation, dating, and analyses.

## 2.3 Ice base determination

The determination of the ice base reflection for this study is composed of existing and newly traced data. We used existing ice base pick data from the AWI MCoRDS 2018/19 campaign from Franke et al. (2021). Ice base determination from the AWI MCoRDS 2023/2024 season was generated using the OPR Toolbox (Open Polar Radar, 2023) and Emerson Paradigm. For the EMR data, we were able to partially rely on existing data (Steinhage et al., 2001; Riedel et al., 2012), which were either directly traced in the 60 ns short-pulse data or, if flown in toggle mode, traced in the 600 ns long-pulse data and projected to the 60 ns data. For EMR short-pulse profiles used in this study that did not have existing ice base picks, the ice base reflection was traced at locations where IRHs were present and the base reflector was clearly visible.





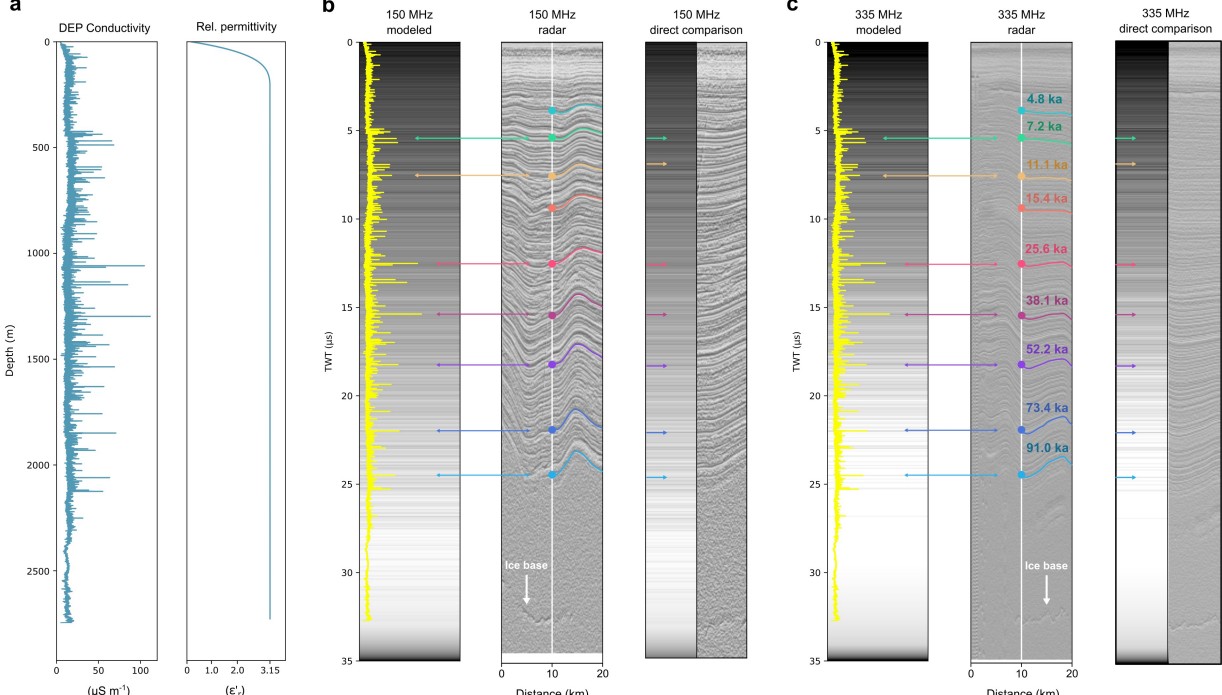

**Figure 2.** Comparison between simulated radar data for 150 and 335 MHz based on measured conductivity at the EDML ice core and radar data collected near EDML. (a) Input parameters for dielectric conductivity and relative permittivity. (b) Comparison between the gprMax simulated radar trace at 150 MHz with overlaid conductivity peaks (yellow) and the EMR profile 20023154 (transmit frequency is 150 MHz). (c) Comparison between the gprMax simulated radar trace at 335 MHz with overlaid conductivity peaks and the MCoRDS wideband profile 20231211_01_024 (centre frequency is 335 MHz). The correspondence between simulated reflections, conductivity peaks, and the reflections of our IRHs is marked with coloured arrows. Note that the simulated radargram is the 2D representation of a single trace (without AGC) and the measured radargram has an along-track range of 20 km (with AGC) to enable a better comparison.

## 2.4 Internal reflection horizon dating

To determine the age of the IRHs, we used two different methods: (1) Age dating by converting the IRH radar wave two-
way travel time (TWT) from the ice surface to depth and correlating it with the age–depth scale of the EDML ice core (e.g., Cavitte et al., 2016; Bodart et al., 2021), and (2) matching reflections of our IRHs with simulated radar reflections based on conductivity peaks derived from measured DEP (dielectric profiling) data (e.g., Eisen et al., 2003, 2006; Winter et al., 2017). Method (1) has the advantage of allowing the determination of an age for each IRH, albeit with the acceptance of a larger error in depth and age assignment. Method (2) offers greater accuracy and linkage to volcanic events because the conductivity peaks
represent the physical origin of reflections (Eisen et al., 2003), however, this approach is not applicable for all IRHs due to some reflections not rising above background noise in the radar data.



**Table 2.** Overview of IRH key characteristics, such as the mean IRH TWT at EDML, mean IRH depth at EDML derived from TWT-to-depth conversion, IRH depths at EDML derived from matching conductivity peaks, IRH ages based on the TWT-to-depth converted depths, IRH ages based on the depth from matched conductivity peaks, and cumulative length of IRHs (line kilometres). The IRH name contains the TWT at EDML in nanoseconds. All TWT-to-depth converted depth values are relative to the ice surface reflection. The conductivity peak depths are from the DEP data set and refer to the depth of the maximum value of the respective conductivity peak.

| IRH name | Mean TWT at EDML | Mean depth at EDML | Depth of conductivity peak | TWT-to-depth-based age | Conductivity peak-based age | IRH line-km |
|---|---|---|---|---|---|---|
| IRH_EDML_TWT_3989_ns | 3 989 ns | 349 m | | *4.8 ± 0.62 ka* | | 28 178 km |
| IRH_EDML_TWT_5503_ns | 5 503 ns | 477 m | 473.00 m | 7.3 ± 0.76 ka | *7.2 ± 0.04 ka* | 29 555 km |
| IRH_EDML_TWT_7704_ns | 7 704 ns | 662 m | 650.25 m | 11.2 ± 0.93 ka | *11.1 ± 0.03 ka[a]* | 24 846 km |
| IRH_EDML_TWT_9527_ns | 9 527 ns | 816 m | | *15.4 ± 1.19 ka[b]* | | 13 308 km |
| IRH_EDML_TWT_12677_ns | 12 677 ns | 1081 m | 1069.72 m | 25.9 ± 2.08 ka | *25.6 ± 0.03 ka[c]* | 23 559 km |
| IRH_EDML_TWT_15661_ns | 15 661 ns | 1332 m | 1311.08 m | 39.0 ± 2.69 ka | *38.1 ± 0.02 ka[d]* | 28 238 km |
| IRH_EDML_TWT_18470_ns | 18 470 ns | 1569 m | 1551.54 m | 53.5 ± 4.22 ka | *52.2 ± 0.07 ka[e]* | 19 277 km |
| IRH_EDML_TWT_22217_ns | 22 217 ns | 1884 m | 1867.55 m | 74.7 ± 5.34 ka | *73.4 ± 0.88 ka[f]* | 25 361 km |
| IRH_EDML_TWT_24723_ns | 24 723 ns | 2095 m | 2080.22 m | 92.4 ± 7.52 ka | *91.0 ± 1.10 ka* | 1 439 km |

[a] Large northern hemisphere higher latitude eruption (rank 18 in Lin et al., 2022)

[b] Large lower latitude or southern hemisphere eruption (rank 32 in Lin et al., 2022)

[c] Taupo Oruanui eruption (Dunbar et al., 2017)

[d] Large lower latitude eruption (rank 3 in Lin et al., 2022)

[e] Large bipolar eruption (rank 5 in Lin et al., 2022)

[f] Toba eruption (Svensson et al., 2013)

### 2.4.1 IRH dating by TWT-to-depth conversion

For the IRH dating using the AICC2023 age–depth scale of the EDML ice core (Bouchet et al., 2023), we used three radar profiles from the seasons 1997/98, 2001/02, and 2023/24, which are located close to the drilling site. Based on the radar traces closest to the ice core for each radar profile (25 m for profile 19983101, 65 m for profile 20023154, and 280 m for profile 20231211_01_024), we determined the travel time between the ice surface reflection and the IRHs. For the depth conversion at EDML, we used the constant radar wave speed in ice suggested by Eisen et al. (2006) of $\sim 1.69 \cdot 10^8$ m s$^{-1}$, corresponding to a dielectric permittivity of $\varepsilon_r' = 3.145$. Additionally, we added a value of 13 m to the IRH depths as a firn correction, calculated from measurements of the complex permittivity of shallow firn cores in the region (Steinhage, 2001; Steinhage et al., 2001). To account for the variation in acquisition years and radar products, we averaged the depth values for all three radar profiles. We disregarded the snow accumulation between the recording times of the radar profiles, as the average accumulation rate at



EDML is approximately 5 cm ice equivalent per year, which amounts to about 1.25 m over 25 years and is significantly below the range resolution of 5 m of the EMR system.

For the depth determination error, we included the following parameters: (1) the standard deviation of the depth differences of the respective IRHs from the three radar profiles, (2) the smallest range resolution of 5 m from the EMR system, (3) the span of the tracing window of 5 ns, which corresponds to approximately 0.5 m, and (4) an error in the dielectric permittivity of 1 % (Bohleber et al., 2012). In addition to the age error resulting from the depth error range, the uncertainty of the age in the AICC2023 chronology is also included.

### 2.4.2   IRH dating by DEP-based radar forward modelling

For age assignment based on simulated radar reflections derived from the measured conductivity of the EDML ice core (Eisen et al., 2006; Winter et al., 2017; Mojtabavi et al., 2022), we used the open-source software gprMax (Giannopoulos, 2005; Warren et al., 2016) to simulate electromagnetic waves in three-dimensional space. This software solves Maxwell's equations using the FDTD method (Taflove and Hagness, 2005) and Yee cells (Yee, 1966), which has proven effective for 1-D and 2-D simulations in ice sheets and glaciers (Franke et al., 2023b; Santin et al., 2023). gprMax uses the same algorithm as already successfully employed for ice-core based forward modelling by Eisen et al. (2006) but includes an efficient solution for model boundary conditions using perfectly matched layers (PMLs; Giannopoulos, 2012) while still offering reasonable computation times for 2-D models. All simulations presented here were performed with gprMax version 3.1.5 (Big Smoke).

For our simulations, we defined a 2-D model domain extending 2.4 m in the x-direction (width) and 2774 m in the z-direction (depth) to represent the full depth of the ice core. To reduce the model size, the y-direction extension was limited to a single cell size, rendering the model 2D to effectively operate in the transverse-electric mode. The cell size in all dimensions is 0.02 m, which enables high resolution, numerical stability, and avoids numerically induced dispersion by ensuring the cell size is at least ten times smaller than the smallest wavelength in our model. Consequently, the time step $\Delta t$ is 0.047 ns for our cell size. The transmit and receive antennas are placed 1 m below the upper limit of the model domain, in the centre of the x-domain (zero offset). The simulation window for all simulations is 30 $\mu$s, covering the entire ice thickness at EDML. The antenna represents a Hertzian dipole transmitting a Ricker wavelet polarized in the y-direction. For all simulations, we defined 30 PML cells at the outer boundary of the x- and z-directions.

We conducted simulations using two fixed transmission frequencies: 150, and 335 MHz. These frequencies represent the centre frequencies of the two radar profiles located close to the ice core (150 MHz for the EMR profile, and 335 MHz for the MCoRDS wideband profile). As input data, we used the conductivity measured by dielectric profiling (DEP; Moore, 1993; Wilhelms et al., 1998) along the EDML ice core. We used a constant dielectric permittivity value of $\varepsilon'_r = 3.145$ for the depth range of solid ice. This corresponds to an EM wave propagation velocity of $\sim 1.69 \cdot 10^8$ m s$^{-1}$, which is the average value determined by Eisen et al. (2006) at EDML. For the upper $\sim 180$ m, we fitted a function to the permittivity measured with the DEP until it reached a constant value. The relative magnetic permeability was set to a constant value of $\mu_r = 1$, and the magnetic loss factor was set to $\sigma_* = 0$.



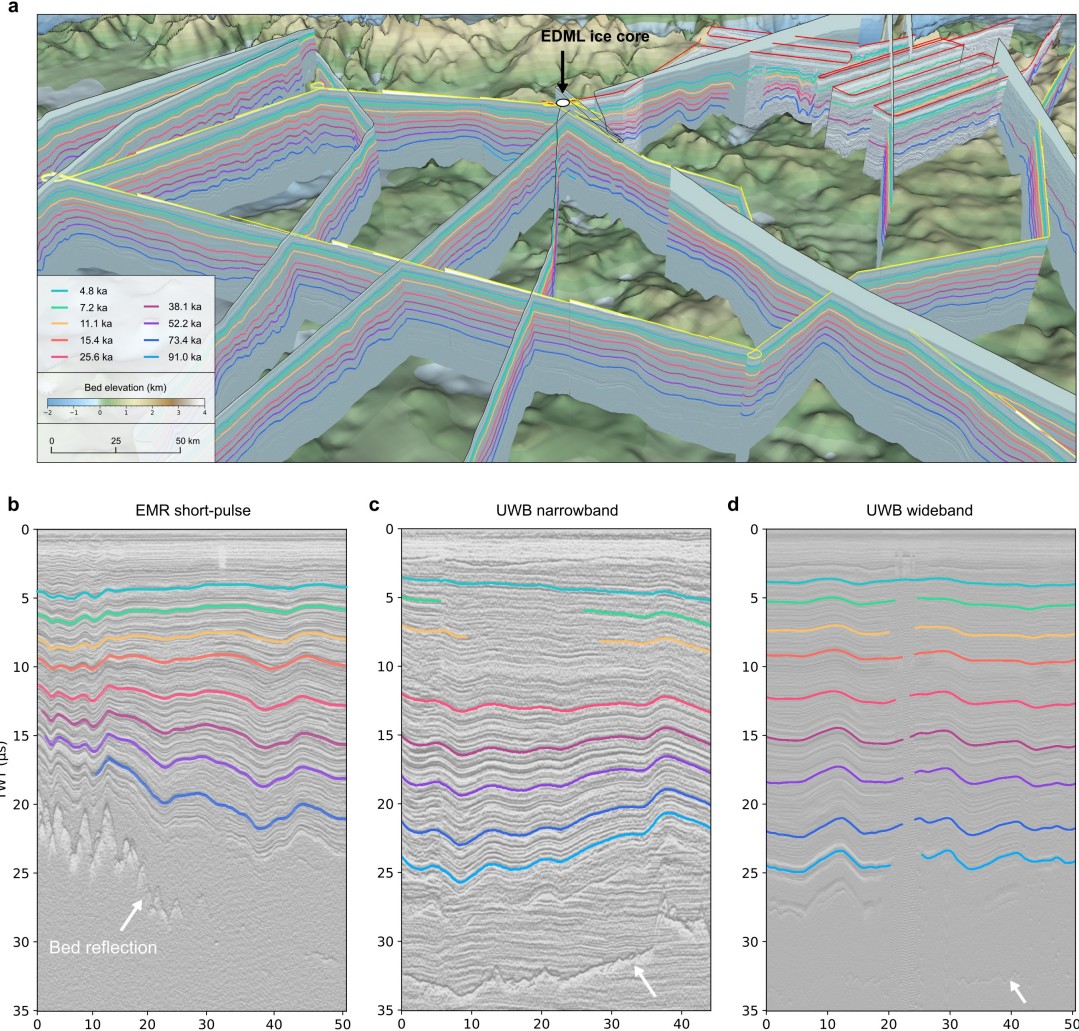

**Figure 3.** Representation of a subset of radargrams (AGC version) of the three different radar products used in this study and traced IRHs in the western DML. (a) Visualization of selected radar data in a 3D canvas with the bed topography (Morlighem et al., 2020) and traced IRHs. Radar profiles with a black line at the top of the fence radargrams are EMR short-pulse profiles, with a red line are narrow bandwidth profiles, and with a yellow line are wide bandwidth profiles. (b) Representation of an EMR short-pulse radargram (profile id: 20023154), (c) a narrow bandwidth radargram (profile id: 20181227_03_001), and (d) a wide bandwidth radargram (profile id: 20231211_01_024) with IRHs. The radargrams in (b-d) are located close to the EDML ice core and are aligned to the surface reflection.

We applied the following signal-processing steps to the simulation output. We derived the reflected energy $P$ of the synthetic radar data following the methods described by Winter et al. (2017) and Franke et al. (2023b). This involved calculating the envelope of the dielectric field strength (y-component of the electric field, $E_y$) using the Hilbert magnitude transform. Addi-



tionally, the radar trace was smoothed using a 1-D Gaussian filter. This approach enhances the representation of the impulse
response of a transmitted radar signal and reflections from a smooth reflector. The final signal strength was obtained by con-

175 verting the electric field envelope to decibel (dB) ($20 log_{10}(Ey)$). We did not apply corrections for geometric spreading or
englacial attenuation.

For the age assignment, we examined which of the simulated reflections originated from conductivity peaks and matched the
IRHs (Eisen et al., 2006). We then determined the depth of the corresponding conductivity peaks (if present and unambiguous to
determine), and used this depth to assign the age via the AICC2023 age–depth chronology (Bouchet et al., 2023). This method

is less dependent on the choice of radar wave propagation velocity in ice, firn correction factor, and ice surface determination,
relying solely on the assignment of conductivity peaks, and provides a clearer age determination with a smaller associated
error. Hence, for the error estimation, we only consider the dating error provided in the AICC2023 age–depth chronology. In
addition, it can be used to better estimate the average relative permittivity of the ice at the ice core site.

## 2.5 IRH depth and normalized depth

We derived the depth of all IRH points by calculating the TWT between the ice surface and the IRH. Then, we converted the
TWT to depth using (e.g., Steinhage et al., 2001; MacGregor et al., 2015; Cavitte et al., 2016; Winter et al., 2017; Ashmore
et al., 2020; Bodart et al., 2021):

$$Z_{irh} = \frac{TWT\, C_{air}}{2\sqrt{\varepsilon'_r}} + Z_{firn}, \qquad (1)$$

where $C_{air}$ is the electromagnetic wave speed in air ($C_{air}$ = 2.9971 $10^8$ m s$^{-1}$). For the calculation of IRH depths, we use

a constant dielectric permittivity of $\varepsilon'_r$ = 3.15. This value deviates slightly from the empirically determined value at EDML
(Eisen et al., 2006), but it is a realistic estimate for the spatial extrapolation within our study area. Additionally, we apply a firn
correction of 13 m (Steinhage, 2001), which is the best-known value in this region.

Additionally, we calculated the normalized depth of the IRHs within the ice column, which provides a better comparison
to determine whether IRHs lie deeper or shallower within the ice column in specific areas. We divided the IRH depth below

195 the ice surface by the ice thickness. Here, we only use ice thickness values at locations where ice base picks are available, as
differences in ice thickness compared to gridded products like BedMachine (Morlighem et al., 2020) can result in significant
discrepancies, which can strongly influence the relative position of IRHs within the ice column. These discrepancies are due to
interpolation methods and the fact that not all ice thickness data used here are yet included in BedMachine.

## 2.6 Data validation

We validated our traced IRHs using a crosspoint analysis for each IRH. To achieve this, we calculated crossover points along
the traced horizons and determined the depth difference between two intersecting IRHs (see Appendix A for details). This
ensured that the same IRHs were traced across intersecting transects across our study area. In total, we analysed more than
2000 crossover points and the differences for all IRHs in the crossover analysis range between 0 and 74 m, with a mean





**Figure 4.** Location, vertical overlap, and spatial distribution of IRHs. (a) Spatial distribution of IRHs from this study (dark blue) in DML and of IRHs from other studies in this region (Winter et al., 2019; Wang et al., 2023, orange and purple, respectively). (b) Color-coded overlap of IRHs. (c – e) Magnified views showing the spatial coverage of IRHs in key regions: (c) Jutulstraumen Glacier onset, (d) EDML ice core site and (e) ice divide south-west of EDML (locations indicated by white boxes in panel b).

standard deviation of 3.75 m. Since we traced the ice surface at the highest gradient of surface return power, while the IRHs

were traced at their maximum return power, a certain fluctuation in depth at the crosspoints is expected. The individual results of the crosspoint analysis and details on the method are presented in Appendix A.





## 3 Results

### 3.1 IRHs ages in EMR short-pulse and MCoRDS data

Two different dating methods provided IRH ages ranging from 4.8 to 91.0 ka (Table 2) covering the Holocene and Last Glacial
Period. We use the age derived from the depth of conductivity peaks to date the IRHs due to their greater accuracy and only use
the age from TWT-depth conversion in the radar profiles when no conductivity peak could be assigned. The radar reflections
of seven of nine IRHs could be clearly associated with conductivity peaks in the EDML core using radar forward modelling
(Fig. 2).

### 3.2 Spatial extent of IRHs in DML

The overall coverage of traceable IRHs in this study spans the entire western DML, covering an area of approximately
450 000 km$^2$. IRHs extend over 700 km south of the EDML ice core, $\sim$200 km north and $\sim$400 km south-west and north-
east. IRHs also extend along the divide between EDML and Dome Fuji ice core sites into a region where IRHs were traced
extensively in AWI's EMR long-pulse data (Fig. 4 a; Wang et al., 2023). The horizontal density of traced IRHs varies signif-
icantly (Fig. 4 c – e) and is highest within a radius of about 30 km around the EDML ice core (Fig. 4 d). Additionally, there
is dense coverage in the Jutulstraumen drainage basin south of the Maud Belt, particularly at the onset of the Jutulstraumen
Ice Stream (Fig. 4 c). Furthermore, there is a high density of IRHs southwest of the EDML ice core at the divide between the
Slessor, Bailey, and Recovery drainage basins and the Riiser-Larsen, Brunt drainage basin (Fig. 4 e). Generally, IRHs cover
mostly slow-moving regions, in particular at the ice divides. Regions of higher flow velocities are covered at the onset of
Slessor ice stream ($\sim$35 m yr$^{-1}$) and Jutulstraumen Glacier ($<$100 m yr$^{-1}$).

IRHs could be consistently traced in most radar profiles. However, there are gaps, particularly in the area of the Jutulstraumen
Ice Stream, which increase downstream as the internal layers become more folded due to ice flow convergence and the deep
canyon system (Franke et al., 2021). Strongly dipping internal layers show a loss in return power depending on their dip angle
at depth due to off-nadir ray path losses and destructive interference (Holschuh et al., 2014). Moreover, gaps in the 4.8 and
sometimes in the 7.2 ka IRH occur because the surface multiple reflection overlays these IRHs, which makes continuous tracing
difficult.

The IRH overlap is greatest in the vicinity of the EDML ice core, where all nine IRHs could be traced nearly continuously.
Here, the layer continuity is least disturbed by ice flow and both shallow and deep IRHs are well-resolved in all radar systems.
There is also high IRH overlap at the onset of the Jutulstraumen Ice Stream and along the ice divide towards Dome Fuji. The
lowest overlap is generally found in the part of radar profiles closest to the Maud Belt, where deeper and older IRHs become
abscent.

In our region, IRHs with ages of 38.1 ka and 73.4 ka were previously traced and published by Winter et al. (2019). This IRH
archive is now significantly expanded by several hundreds of kilometres around EDML. The 38.1 ka and 73.4 ka horizons are
extended by $\sim$450 % and $\sim$480 %, respectively. Additionally, seven more IRHs were added with comparable coverage (Figure





4 b). Based on the existing dataset of IRHs from (Winter et al., 2019), this represents a total increase in IRH data points by ∼ 1860 %.

## 3.3 Spatial variation in IRH depth distribution

Our IRHs, that range between 4.8 and 91.0 ka in age, are found at depths between 200 and 2200 meters below the ice surface (Figure 5). Both the absolute depth below the ice surface and normalized depth distribution of the IRHs show significant variation. We observe a general pattern where IRHs around the ice divide near EDML and to the west are deeper compared to IRHs further south and towards Dome Fuji (Figure 5 and 6). This pattern is even more pronounced in the normalized IRH depths along the ice divide between EDML and Dome Fuji. Additionally, we note a contrasting trend when comparing absolute and relative IRH depths at the edges of the Maud Belt and on the plateau. For the 73.4 ka IRH, for example, it is evident that northeast of EDML, this IRH lies at depths of approximately 1000 to 1200 meters, similar to the southeastern part near Dome Fuji (comparison of Feature A and Feature B in Figure 5). However, the relative depth reveals that the 73.4 ka IRH is about 65 % below the surface northeast of EDML, but only about 45 % near Dome Fuji. This pattern is also evident in the 11.1 ka, 25.6 ka and 38.1 ka IRHs, however with smaller differences as the IRHs become younger (comparison of Feature A and Feature B in Figure 6).

The IRH depths within the Jutulstraumen drainage basin are particularly variable and show the following pattern in their relative depths: In the southwestern part of the Jutulstraumen onset region, the IRHs are significantly deeper (absolute depth) than in the northeastern area. The IRHs in this region (denoted as Feature A in Figures 5 and 6) are significantly shallower compared to the southwest and east but show similar relative depths to EDML. This general pattern is evident in all IRHs that cover this region. A different pattern emerges when considering absolute depths. We find a greater variation with very shallow IRHs in the northeast and very deep IRHs in the Jutulstraumen Trough area. In the 38.1 ka IRH, these depth differences amount to up to 1500 m.

Additionally, the variation in depth of the IRHs is striking in two profiles extending from EDML towards the south-southeast. Around 79.9° S, there is a significant increase in absolute depth followed by a decrease to more shallow depths, observable in all IRHs from 7.2 ka onward (Feature C in Figure 5). This pattern is more pronounced in older IRHs compared to younger IRHs. However, there is no information on the relative IRH depth at this location since the ice base reflection is not visible in this section of the radar profile. Moreover, we observe an increase in absolute IRH depth at the southwestern edge of our IRH data, which is particularly pronounced in the 25.6 and  ka and 38.1 ka IRHs (Feature D in Figure 5). Notably, older IRHs of this particular region could not be traced in the radar data.







**Figure 5.** Color-coded absolute depth representation of the nine IRHs from this study. The background map displays the ice surface elevation REMA (Howat et al., 2019) in hillshade. The fine black lines delineate the drainage basins according to Zwally et al. (2012).



**Figure 6.** Color-coded normalized depth representation of the nine IRHs from this study. The background map displays the ice surface elevation REMA (Howat et al., 2019) in hillshade. The fine black lines delineate the drainage basins according to Zwally et al. (2012).

## 4 Discussion

### 4.1 Interpretation of IRH depth distribution variations

Feature A (Figures 5 and 6) represents a notable anomaly in a region characterized by particularly shallow IRHs in both
absolute and relative depths. This region deviates from the general trend of having a gradient from shallower IRHs on the



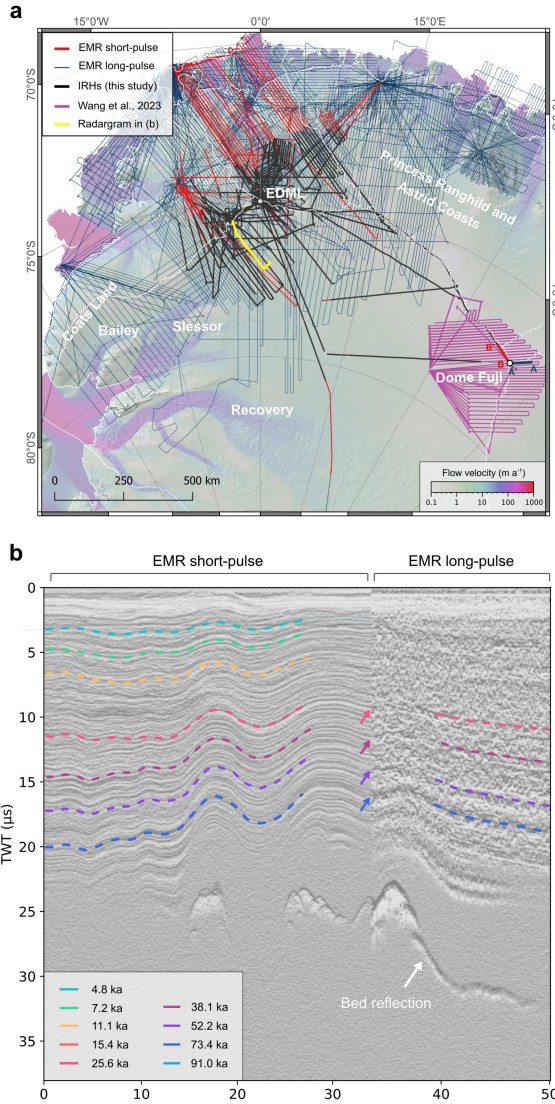

**Figure 7.** Potential to extrapolate IRHs to AWI EMR long-pulse data. (a) Overview of EMR short-pulse data (red) superimposed with the IRHs traced in this study (black), and in the background showing the EMR long-pulse data (dark blue). The IRHs from Wang et al. (2023) are depicted in purple. The lines A – A' (black) and B – B' (red) in a) indicate a long-pulse and short-pulse radargram at Dome Fuji (Figure B1) where IRHs from this study and IRHs from Wang et al. (2023) intersect. (b) Composite radargram (AGC version) from EMR short-pulse and long-pulse data (profile ID 19992116, long-pulse, right; 19993116, short-pulse, left). At the intersection, transitions where IRHs from this study are also represented in the long-pulse data are marked with arrows.

plateau to deeper IRHs (especially in relative depth) near the Maud Belt. This general trend is likely linked to the overall snow accumulation pattern, with higher accumulation rate towards the Antarctic coast and lower accumulation rate on the



plateau. The shallow IRHs in Feature A are likely also accumulation-driven, as this area shows significantly reduced snow accumulation, as determined from firn core analysis (Rotschky et al., 2007). The fact that this pattern is also evident in older IRHs suggests that this low-accumulation anomaly may have persisted in this region not only during the Holocene but also during the Last Glacial Period, given that the ice in this region shows very low flow velocities.

Additionally, IRHs are particularly deep (both absolute and relative) in the southwestern area of the Jutulstraumen onset, which could be linked to the ice stream activity. Besides dynamic thinning, increased friction at the ice base could lead to basal melting, causing the IRHs to subside both absolutely and relatively. However, our work also shows that ice older than 73.4 ka is present in the main trunk of Jutulstraumen, at least at its onset. Further northeast, the shallow depths of the IRHs might be explained not only by the low-accumulation anomaly but also by reduced basal sliding, less basal melting, and even a positive basal mass balance due to basal freeze-on on subglacial water (Franke et al., 2021, 2024).

A comparison of IRH depths between Feature A (south of the Maud Belt) and Feature B (near Dome Fuji) highlights the importance of considering the relative depth of IRHs in relation to ice thickness. The fact that all IRHs traced near Dome Fuji are significantly shallower relative to ice thickness than those further northwest can be explained by the ice sheet's geometry and surface mass balance. Both regions are characterized by low accumulation rates (Rotschky et al., 2007; Oyabu et al., 2023), but the ice at Dome Fuji is overall much thicker and significantly older at greater depths compared to further north. This results in lower relative depths for the IRHs.

The sudden IRH depth increase of all ages along the southward-trending radar profiles (Feature C) and some IRHs in the southwest (Feature D) could be explained by changes in bed elevation. Although there are no visible bed reflections in the radar profiles where the IRHs were traced, ice thickness data from nearby regions suggest a connection with the underlying topography. The bed topography data from BedMachine (Morlighem et al., 2020) indicate that the IRH depth increase in Feature C is associated with a subglacial valley with a width between $15 - 25$ km. The IRHs at Feature D are located at the onset of one of the tributaries of the Slessor Ice Stream. Higher flow velocity at this point would dynamically thin the ice, which must be compensated by lateral mass influx or increased snow accumulation, which would increase IRH depths. Hence, the relative drop of the IRHs in this area could indicate the long-term activity of this tributary. The fact that older IRHs, such as the 52.2 ka and 73.4 ka reflector, could not be traced in this area further supports the hypothesis of a significant dynamic component causing IRH subsidence or layer disruption, which could suggest that this ice stream system has been stable at least for the past $\sim 70$ ka.

## 4.2 Link between IRHs and volcanic eruptions

The dominant origin of IRHs was attributed to changes in conductivity due to variations in acidity originating from deposits of large volcanic events (Millar, 1981). Six of the IRHs that we dated using conductivity peaks and one depth-based IRH can be associated with major bipolar volcanic eruptions during the Holocene and Last Glacial Period (Lin et al., 2022; Svensson et al., 2013). Our 11.1 ka IRH, associated with a moderate conductivity peak, can be correlated with the deposits of a bipolar eruption event dated to 11.3 ka BP. These deposits have been identified in both Antarctic and Greenland ice cores (Lin et al., 2022). Additionally, we consider the possibility that our 15.4 ka IRH could be linked to an event dated to 15.6 ka BP. Although





there is no clear and strong conductivity peak in the EDML ice core for this event, one explanation could be that this event is classified as a relatively weaker bipolar eruption compared to others (rank 32; Lin et al., 2022).

The radar reflection of the 25.6 ka IRH can be linked to sulfate deposits from the New Zealand Taupo, Oruanui eruption,
25.32 ka BP, which were detected in two Greenland (undetected in GISP2) and three Antarctic ice cores (Lin et al., 2022) and confirmed by tephra deposits in Antarctica (Dunbar et al., 2017). The IRH we dated to 38.1 ka is very likely associated with the third strongest bipolar eruption in the Last Glacial Period known to date, which occurred at 38.13 ka BP. The exact eruption location is unknown, but it is likely a low-latitude or Southern Hemisphere eruption with an estimated average climate forcing twice as strong as the Tambora eruption in 1815 CE (Lin et al., 2022). Our 52.2 ka IRH could correspond to an eruption that
occurred at 52.2 ka BP and is listed as the fifth strongest eruption in Lin et al. (2022). The deposits from this eruption can be traced in three Greenland and three Antarctic ice cores (Lin et al., 2022), and the possible eruption site is estimated to be in the high-latitude Northern Hemisphere (above 40° N).

From tracing IRHs in the radar data in this region, we find that the reflection of the 73.4 ka IRH is the most discernible and extensive in the deeper third of the ice sheet. As per Winter et al. (2019), we also speculate that this reflection most likely
corresponds to the deposits of the Toba eruption in northwest Indonesia, dated using tephra and Ar-Ar dating to $73.88 \pm 0.32$ ka BP and $75.0 \pm 0.9$ ka BP, depending on the location of the deposits (see references in Svensson et al., 2013). The Toba eruption is considered the largest known supereruption of the last 2.5 million years (Chesner, 2012) and likely occurred during the cooling transition from Greenland Interstadial 20 to Greenland Stadial 20 (Lin et al., 2023, and references therein).

### 4.3   Comparison with IRHs from other studies

Steinhage et al. (2013) traced eight IRHs along the ice divide connecting EMR short-pulse profiles between EDML and Dome Fuji and dated them via TWT-to-depth conversion using the EDML AICC2012 age–depth chronology (Bazin et al., 2013). Several of these isochrones correspond with those we have traced over a much larger spatial extent in this study (corresponding IRH ages dated at the EDML core by Steinhage et al., 2013, shown in parentheses): 4.8 ka (4.7 ka), 7.2 ka (7.4 ka), 25.6 ka (25.1 ka), 38.1 ka (38.1 ka), and 73.4 ka (72.4 ka). The discrepancies arise due to a combination of the choice of radar profiles,
different dating methods, the type of surface pick, and the selected propagation speed for radar waves in ice, all of which are within the stated error margins. Additionally, the IRHs in Steinhage et al. (2013) were also dated using the Dome Fuji age–depth chronology, resulting in different ages. Moreover, the 38.1 ka and 73.4 ka IRHs were already traced to a large extent in our study area by Winter et al. (2019). These data provide an essential foundation for this study, as they seamlessly connect with our dataset. Winter et al. (2019) also mention that IRHs with ages of 4.8, 7.6, 15.4, and 25.0 ka were traced to a smaller
extent (and very likely correspond to the same IRHs in our study), but were not published.

A comparison with traced IRHs in East Antarctica, in the region of the Vostok ice core and EPICA Dome C ice core (Leysinger Vieli et al., 2011; Cavitte et al., 2016, 2021; Winter et al., 2019) suggests that some of these IRHs likely correspond to those identified in our study. Most of these isochrones extend far beyond an age of 100 ka, reaching up to approximately 700 ka, with the dating of the oldest IRHs being particularly challenging due to the inherent uncertainties associated with the
1-D age-depth model used to date (Cavitte et al., 2021). The 38.1 ka and 73.4 ka IRHs (dated to the same ages by Cavitte



et al., 2021), are almost certainly the same reflections in this region, as they exhibit overall very strong reflections in our survey area and correlate with significant past volcanic eruptions (Lin et al., 2022; Svensson et al., 2013). Furthermore, our isochrones extend into the area around Dome Fuji, for which there are dated IRHs from Wang et al. (2023). However, these were traced using EMR long-pulse data, which have a vertical resolution of ∼ 50 m, making the direct comparison more difficult.

Nonetheless, they also identified IRHs dated to $36.3 \pm 3.6$ ka and $75.3 \pm 7.0$ ka, which, when examined at the intersection points with our EMR short-pulse radar data, most likely represent the same englacial reflectors (see Appendix B).

Dated IRHs with similar ages to ours are not confined to East Antarctica. Several studies have traced, dated, and published IRHs in West Antarctica (Muldoon et al., 2018; Ashmore et al., 2020; Beem et al., 2021; Bodart et al., 2021, 2023). Notably, in particular, our 4.8 ka IRH appears to be widespread in West Antarctica. For instance, Muldoon et al. (2018) identified a 4.7 ka

IRH across West Antarctica's ice divide and throughout the Thwaites Glacier catchment, Beem et al. (2021) traced a 4.7 ka IRH at Titan Dome, and Bodart et al. (2021, 2023) identified a 4.72 ka IRH across the divide and throughout the Pine Island Glaicer catchment, with direct links showing vertical conformity with another IRH widely traced across the Institute and Möller Ice Stream Catchment by Ashmore et al. (2020). Additionally, there may be further links between our 7.2 ka IRH and the 6.94 ka IRH from Ashmore et al. (2020); Bodart et al. (2021, 2023), our 25.6 ka IRH and the 24.9 ka IRH from Muldoon et al. (2018),

and our 11.1 ka and 73.4 ka IRHs and the 10.7 ka and 72.5 ka IRHs from Beem et al. (2021). However, to accurately determine if these dated isochrones indeed originate from the same reflector Antarctic-wide, it would be necessary to link not only the IRH ages (which can vary depending on the ice core, method, and determined TWT between the ice surface and the IRH) but also the backscatter patterns in the radar data. Here, radar lines connecting datasets between East and West Antarctica would be beneficial. However, this presents challenges, such as long flights over the Antarctic plateau, and geographical difficulties,

such as the loss of layer continuity across the Transantarctic Mountains. For a definitive connection of IRHs that relate to the same reflectors and, therefore, the same deposits, geochemical analyses of ice cores could provide additional constraints.

In the future, the AntArchitecture initiative will aim to compile all such IRH datasets into a continent-wide radiostratigraphic database, which will be used to create a three-dimensional age–depth model of the ice sheet by interpolating between known IRH along-track datasets similar to already existing gridded topographic models such as Bedmap (Lythe and Vaughan, 2001;

Fretwell et al., 2013; Frémand et al., 2022) and BedMachine Antarctica (Morlighem et al., 2020). For such a task to be successful, additional IRH tracing will need to take place in poorly surveyed areas of the ice sheet or away from ice divides and ice-core locations using existing RES coverage to avoid over-fitting the interpolation algorithms. By utilizing three decades of RES data from multiple radar systems in the area, this study bridges a significant gap in the East Antarctic radiostratigraphic record and thus in our ability to obtain a continent-wide age–depth model of the ice sheet, as envisaged by AntArchitecture.

**4.4   Potential to expand IRH coverage in western DML**

In our study, we have not fully utilized the complete archive of AWI radar data. We focused on EMR short-pulse, MCoRDS narrow and MCoRDS wide bandwidth data because these have similar vertical resolutions (5 m or better), enabling the capture of a higher number of IRHs that are comparably represented across these systems. However, there are additional areas in DML covered exclusively by EMR long-pulse data (approximately 50 m vertical resolution but deep-sounding). At intersections





between EMR short-pulse and long-pulse data where we were able to trace IRHs, it becomes apparent that the 25.6, 38.1, 52.2, and 73.4 ka IRHs are also represented in the long-pulse data (Fig. 7 b), which aligns with the IRHs of similar age in Wang et al. (2023). The representation of the same reflectors in EMR long-pulse and short-pulse data does not only account for reflections caused by changes in electrical conductivity but also for reflections caused by changes in ice crystal lattice orientation (Eisen et al., 2007). Considering the AWI EMR radar line coverage in Figure 7 a, there is additional potential,

particularly for regions upstream of Princess Ragnhild and Princess Astrid coasts, as well as the onsets of the Slessor and Recovery ice streams. Additionally, the spatial coverage of the East Antarctic plateau between EDML and Dome Fuji could be significantly improved. It may even be possible to establish connections between our IRHs and the western Coats Land at the ice divide to the Bailey and Slessor basins using AWI's and the British Antarctic Survey's radar data (Frémand et al., 2022), which are of similar range resolution.

## 4.5 Significance of dated IRHs for reconstructing the ice-sheet history in western DML

Reconstructing past ice sheet configurations is crucial for understanding ice-sheet processes and their impacts on the Earth system, as well as providing essential constraints for ice sheet models. The spatial depth distribution of our nine IRHs in western DML can significantly contribute to understanding ice-sheet evolution. By comparing IRH depths and spatial anomalies, we can test hypotheses about underlying mechanisms, such as surface or basal mass balance and ice-dynamic processes, and how

they may have changed over time. Our dated IRHs also provide an opportunity to validate or expand existing findings and hypotheses on ice surface changes based on cosmogenic nuclides (Andersen et al., 2020) about past ice dynamics in DML (Andersen et al., 2023; Braga et al., 2023). In addition, the absolute and relative depth of our IRHs offer the opportunity to complement studies on e.g., preserved paleo-geomorphological structures (e.g., Näslund, 1997; Rose et al., 2015; Franke et al., 2021; Carter et al., 2024) or regions of basal freeze-on of subglacial water (e.g., Bell et al., 2011; Leysinger Vieli et al., 2018;

Franke et al., 2024) providing further insights into past flow behaviour and ice sheet configurations.

Furthermore, we see significant potential for improving regional and Antarctic-wide ice sheet models to provide better projections for future sea-level rise (Sutter et al., 2021). Our IRHs can serve as a crucial tool for calibrating results in model simulations (Sutter et al., 2021; Višnjević et al., 2022). Especially in dynamic regions such as the Slessor and Jutulstraumen ice stream onset regions, our IRHs are particularly valuable for testing hypotheses of their stability over multiple glacial cycles

(Rippin et al., 2003, 2006). Conversely, IRH data from less dynamically active regions, such as the ice divide south-west from EDML, along the divide between EDML and Dome Fuji, the region south of EDML, as well as the shallow-IRH anomaly region (Feature A), are well-suited for inferring the spatial distribution of past accumulation rates (Eisen et al., 2005; Huybrechts et al., 2009; Leysinger Vieli et al., 2011).

An additional approach to decipher the past ice dynamics of a region using IRHs lies in the structural analysis of three-

dimensional englacial structures, particularly the geometries of folds, fold axial planes, and the development of fold amplitudes with depth (Bons et al., 2016; Franke et al., 2023a; Jansen et al., 2024). Especially for Greenland, it has been shown in various dynamic ice regions that a three-dimensional representation aiming to depict deformation structures like folds accurately can contribute to understanding ice mechanical properties (Bons et al., 2016) and past flow patterns (Franke et al., 2022a; Jansen



et al., 2024). Notably, the profiles in the upstream region of the Jutulstraumen Ice Stream, as well as the region around EDML,
where the high density of IRHs allows resolving small-scale englacial structures, and the profiles at the ice divides, have the
potential to provide insights into past ice dynamics, such as ice flow direction changes or shifts in the location of ice divides.

## 5 Conclusions

We have mapped nine internal reflection horizons (IRHs) using radar data from various systems of the Alfred Wegener Institute,
collected over the last three decades. Our IRHs cover the western DML, south of the Maud Belt, and range from 4.8 to 91.0 ka
BP covering the Holocene and Last Glacial Period. Accurate dating of the IRHs was achieved by combining the age–depth
chronology of the EPICA DML ice core and DEP-based radar forward modelling. Additionally, six of the new IRHs could
be linked to deposits from significant historic bipolar volcanic eruptions, facilitating synchronization of these reflectors across
Antarctica and potentially Greenland. Many of the IRHs mapped in this study likely correspond to the same englacial reflectors
found in extensive regions over East and West Antarctica. A comparison with AWI EMR long-pulse data suggests that some
of the IRHs identified here could be extended to even larger sectors of East Antarctica.

Our results significantly contribute to a broader understanding of the englacial age architecture of the DML Region in East
Antarctica and highlight the potential for linking individual IRHs to other regions in Antarctica. Furthermore, the findings
presented here are fundamental for enhancing our comprehension of past ice sheet processes and are crucial for numerical ice
flow models aimed at improving our understanding of the paleoclimatic history of Antarctica.

*Data availability.* Our IRHs will be made publically available at PANGAEA upon publication. Reviewer access to the IRH data is provided
via Nextcloud: https://nextcloud.awi.de/s/DtYZCiiasxbxWHA. The AICC2023, EDML ice core age–depth chronology (Bouchet et al., 2023)
is available at PANGAEA: https://doi.pangaea.de/10.1594/PANGAEA.961019. Ice surface velocities from Mouginot et al. (2019) are avail-
able at the National Snow and Ice Data Center (NSIDC), https://doi.org/10.7280/D10D4Z. The drainage system boundaries Zwally et al.
(2012) can be obtained here: https://earth.gsfc.nasa.gov/cryo/data/polar-altimetry/antarctic-and-greenland-drainage-systems. The BedMa-
chine Antarctica V03 bed topography data from Morlighem et al. (2020) is available at https://nsidc.org/data/nsidc-0756/versions/3. The
Reference Elevation Model of Antarctica (REMA) is available at the Polar Geospatial Centre: https://www.pgc.umn.edu/data/rema/.

## Appendix A: Crosspoint analysis

We performed an analysis of IRH depth differences at crossover points of intersecting radar lines to validate that we assigned
the same reflections for the respective IRHs. For the analysis of crossover differences in the respective IRHs, we calculate the
exact intersection points between the geometric lines formed by connecting the individual IRH picks. We create line subsets of
one radar profile where IRH gaps exist to avoid creating lines, and therefore potential fake intersection points, where no IRHs
are traced within a radar profile. Using these lines, we determine the location of the resulting intersection points. We then create
a circular buffer around these intersection points and capture all IRH picks within this buffer. We select a buffer radius of 50 m



and average the IRH depths of all picks within a profile to minimize small-scale variations introduced by the semi-automatic
tracker. The difference is derived from the mean depths of picks from both intersecting lines within the buffer.

Examining the histograms in Figure A2, we observe that most intersection point differences are below 10 meters depth
difference, and the depth difference generally decreases for higher values. For the 4.8 ka and 7.2 ka IRHs, we see a small local
maximum around a 5 m intersection point difference. Moreover, the differences become larger for older IRHs, leading to a
higher number of larger crosspoint errors compared to younger IRHs.

## Appendix B: Connection to Dome Fuji IRHs

We can ascertain several points from a comparison of our IRHs with those from Wang et al. (2023), who traced IRHs around
Dome Fuji using EMR long-pulse data. The IRHs from Wang et al. (2023) begin at greater depths relative to the surface,
making them older relative to the IRHs in our study. However, the Dome Fuji region also allows tracing much older IRHs (up
to 230 ka BP). Due to the availability of only EMR long-pulse data in this region, the range resolution is significantly coarser
(approximately 50 m), resulting in a larger error in age dating compared to our IRHs. A direct comparison of two radargrams
intersecting at Dome Fuji confirms that both the 36.3 ka and 75.3 ka IRHs from Wang et al. (2023) correspond to the same
reflectors identified in our study, which we date as 38.1 ka and 73.4 ka, respectively (Figure B1). Additionally, it is likely that
the reflection of the 135.6 ka IRH from Wang et al. (2023) corresponds to a reflection observed in the EMR short-pulse data
(Figure B1).

## Appendix C: IRH data set description

We publish our dated IRHs on the *PANGAEA* Data Publisher (Felden et al., 2023) and comprises the information summarized
in Table C1. It is important to note that the vertical position of the IRHs is provided in different formats: (1) as absolute two-
way travel time (TWT) in the radargram, representing the direct reference to the radargram; (2) as TWT beneath our ice surface
pick. To ensure this reference is clear for future users, we also provide the absolute TWT of the ice surface pick. For conversion
between TWT to depth, we used a dielectric permittivity of $\varepsilon'_r = 3.15$ and a firn correction term of $Z_{firn} = 13$ m (see Eq. 1).
The IRH depth and elevation are referenced to the REMA surface DEM (version 2; Howat et al., 2019), relative to the WGS84
ellipsoid. The Profile ID is the ID of the radargram, which also appears in the filename. The Paradigm ID is an internal AWI
ID for referencing the AWI radar database and radar processing software where the picks are created and archived.

*Author contributions.* SF conceptualised the study, wrote the manuscript, traced and dated all IRHs, performed all data analysis, created all
figures and validated the data. AMZ, DS and SF acquired radar data from the 2023/24 AWI MCoRDS campaign, while AMZ processed
the data from this campaign with contributions of SF. AMZ, VH and SF contributed to tracing the ice-bed interface. JB provided context of
comparable studies and for the discussion about connecting IRHs in this study over Antarctica. VH developed an algorithm for standard ice







**Figure A1.** Differences of IRHs up to 50 m at crossing points.

surface tracking and processed GPS data. OE, DJ and VH implemented access and usability to AWI's radar data archive. All authors jointly revised and edited the manuscript.



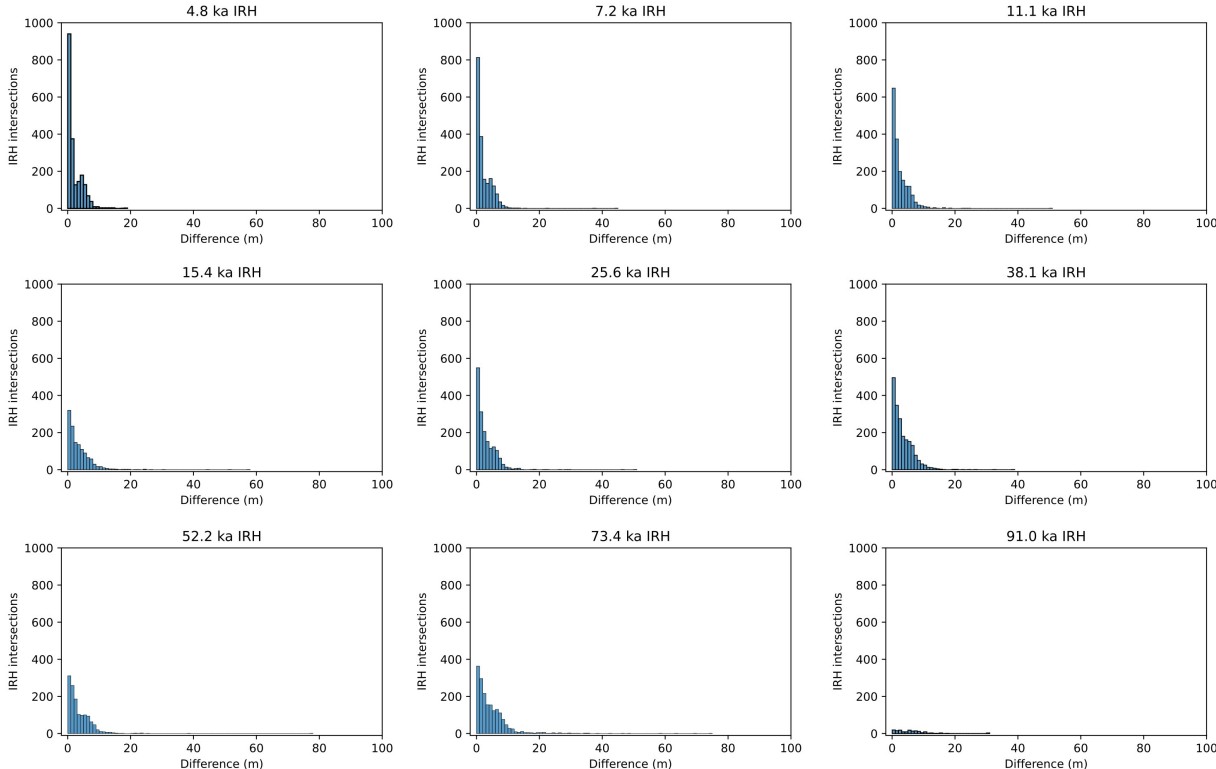

**Figure A2.** Histograms of the total count of differences of IRHs at crossing points up to 100 m. The bin width is 1 m.

*Competing interests.* The authors declare no competing interests.

*Acknowledgements.* We thank the AWI polar aircraft technicians Eduard Gebhard and Christoph Petersen for their support in the field during the 2023/24 radar campaign, the Kenn Borek crew of AWI's polar research aircrafts, as well as all supporters of previous radar campaigns mentioned in acknowledgements of related publications. Logistical support in the field over the past three decades has been provided by Neumayer III Station (Germany), Troll Station (Norway), Kohnen Station (Germany), Princess Elisabeth Station (Belgium),

and Novolazarevskaya Airbase (Russia). We acknowledge the use of software from Open Polar Radar generated with support from the University of Kansas, NASA grants 80NSSC20K1242 and 80NSSC21K0753, and NSF grants OPP-2027615, OPP-2019719, OPP-1739003, IIS-1838230, RISE-2126503, RISE-2127606, and RISE-2126468. The authors would like to thank Aspen Technology, Inc. for providing software licenses and support. This study was motivated by the AntArchitecture Scientific Committee on Antarctic Research Action Group.

Steven Franke was funded by the Walter Benjamin Programme of the Deutsche Forschungsgemeinschaft (DFG, German Research Founda-

tion; project number 506043073). Alexandra M. Zuhr was funded by the DFG in the framework of the priority program SPP 1158 "Antarctic Research with comparative investigations in Arctic ice areas" (grant number 522419679). Julien A. Bodart acknowledges funding from the Swiss National Science Foundation (grant number 211542).



**Figure B1.** Illustration of two radargrams (AGC version) and the IRHs intersecting at the Dome Fuji ice core. Panel (a) shows the radargrams with IRHs, and panel (b) shows the radargrams without IRHs but where the Wang et al. (2023) IRHs intersect this study and Winter et al. (2019)'s IRHs at the Dome Fuji ice-core site. (a) The radargram on the left is an EMR long-pulse profile with IRHs from Wang et al. (2023), while the radargram on the right is an EMR short-pulse profile with IRHs from this study together with the IRHs from Winter et al. (2019). (b) The location at the Dome Fuji core, where the 36.3 ka and 75.3 ka IRH from Wang et al. (2023) intersects with the 38.1 ka and 73.4 ka IRH from this study and Winter et al. (2019) are marked with arrows. The colours of the IRHs in the left radargram correspond to those in Figure 2 in Wang et al. (2023). The location of the two profiles is shown in Figure 7 a.



**Table C1.** IRH data set description for column-separated text files.

| Column name | Unit | Description |
| --- | --- | --- |
| Profile ID | | Radar profile ID |
| Paradigm ID | | Radar Profile ID in the Paradigm system (AWI picking software) |
| Radar product | | EMR short-pulse, MCoRDS narrowband, or MCoRDS wideband |
| Longitude | decimal degree | EPSG:4326 |
| Latitude | decimal degree | EPSG:4326 |
| IRH TWT | ns | absolute TWT in nanoseconds |
| IRH TWT ice | ns | TWT below ice surface in nanoseconds |
| Surface TWT | ns | TWT to ice surface in nanoseconds |
| Base TWT [*] | ns | TWT to ice base in nanoseconds |
| IRH depth | m | |

[*] The ice base reflection is not present in all radar profiles.

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
