# Peer review of "Age-depth distribution in western Dronning Maud Land, East Antarctica, from three decades of radar surveys"

_EGUsphere, 2024_

## Author Response (AR1)

**Public justification (visible to the public if the article is accepted and published):**
Hi Dr. Franke et al.,

Based on the overall quality of the submitted MS, our discussion regarding the MS's suitability for The Cryosphere (vs. e.g., ESSD), the referees' insightful comments that presented similar arguments, and the quality of your response to those comments, I expect that a revised MS may be suitable for publication in The Cryosphere. However, because of the expected expansion of its scope to delve more into the dataset's scientific implications, I will ask referees to take another look at it (as well as myself).

Regards,

Joe MacGregor
NASA/GSFC

Dear Joe MacGregor,

Thank you very much for handling our manuscript submitted to *The Cryosphere*. We have thoroughly addressed the reviewers' comments and suggestions and believe that our revised manuscript is now in a much-improved form.

We have expanded the scope of the study by:

1. Providing a more detailed and specific discussion on the IRH depths in relation to glaciologically relevant aspects, such as bed topography, accumulation rates, and ice flow velocity.

2. Enhancing the linkage between our IRHs and those from other studies, now illustrated with a new figure and a detailed table.

The responses to the reviewers' comments can be found further in the document.

Best regards,
Steven Franke

**RC1**: 'Comment on egusphere-2024-2349', Rebecca Sanderson, 18 Sep 2024

**Review of "Age-depth distribution in western Dronning Maud Land, East Antarctica, from three decades of radar surveys"**

The research uses radio echo sounding data to explore internal reflection horizons (IRHs) and the age-depth distribution of ice across western Dronning Maud Land, East Antarctica. Using various radar systems, the researchers trace nine IRHs and assign ages from a two-way travel time to depth conversion and by employing radar forward modelling based on conductivity peaks from the DML ice core. The article presents a new, useful cryospheric dataset that has been used here to explore the englacial architecture across DML. The work is very well written, meticulously constructed, clearly elucidates all the methodologies employed, as well as demonstrating the possibilities of tracing the same IRH across multiple radar surveys.

While the article presents a very useful data set, I believe that its application could be expanded in places, e.g. the relationship of the IRH depths/normalised depth with current ice velocity or accumulation rates (may only require a figure and a line or two adding in places). This would allow for a science focus rather than a data driven focus. I believe that this can be achieved through minor corrections, as the data is already telling the story. Therefore, I suggested that the paper be published with only minor revisions to account for this.

> We would like to thank Rebecca Sanderson for taking the time to review our paper and we appreciate the constructive suggestions and the overall positive feedback. Below, we address the key aspects raised.

> As suggested by the reviewer, we have expanded Section 4.5 to deepen the discussion of the direct implications of IRH depths (or normalized depths) by comparing them to ice flow velocity, bed topography, and accumulation rates. Additionally, we have included a new figure to support this discussion.

**Specific comments:**

In the first paragraph of 3.2, you need to be explicit that you are talking about traceable IRHs.

> We are not sure if we got the reviewer's point correctly, because the first sentence of this paragraph starts with: *"The overall coverage of **traceable** IRHs in this study spans the entire western DML, covering an area of [...]".* We therefore do not see a need to make further changes here.

In section 4.2 you link IRH to volcanic eruptions, this is really interesting for the broader picture but for most of the IRH ages, the suggested eruption dates are a couple of

hundred years out, either before or after the eruption. Could you explain the reason for this, e.g. do you think this just a function of dating uncertainty/ radar system resolution?

> We assume that the discrepancies in age between the IRH dating and the ages of volcanic eruptions, whose deposits make the IRHs visible in the radargram, are caused by a combination of the respective dating methods. The example of the dating of the Toba eruption (Svensson et al., 2013) shows that the dating method of volcanic eruptions is crucial. Although we obtain a very precise dating based on the depths of the assigned conductivity peaks at the EDML core, this age refers only to that specific ice core. It would not surprise us if the dating of IRHs at other cores, using the same method, produced different results, yet still had a small age error at each individual core.

> As suggested by the reviewer, we have now included this aspect in the respective section of our manuscript.

I think it would be useful to expand section 4.5 to talk about some of the direct application of your dataset. I think the addition of a more detailed explanation of how the IRH that you have traced in the DML region represent dynamic changes over time would improve the manuscript. This could include links to ice velocity, bed topography, accumulation rates.

> We agree with the reviewer and expanded Section 4.5 to include more direct applications. In line with one of the earlier comments, we have now better illustrated the direct comparison of IRH depths (or normalized depths) with ice velocity, bed elevation, and accumulation rates, as well as the implications for dynamic changes (new Figure 7).

Your figures are clear and support the text well, but I found myself jumping around a lot when they are mentioned in the text. For example figure 2 is on page 6 but the first time you refer to it is page 12. Likewise, I am unsure that you refer to figure 3 at all in the text.

> Thank you for spotting this, we now reference Figure 3. We will rearrange the figure position in the text to be in line with the content, however, the final decision will be made by the typesetting team.

**Line suggestions:**

Line 61 Refer to figure 1 here.

> Done.

Line 105 "An additional gain function was applied to the radar data (also known internally as AGG products) to enhance reflections at all depths." Unnecessary repetition of text at the beginning of the paragraph, delete this?

Done.

Figure 2 caption. Please add the following to be clear as to why you used the frequencies: "Comparison between simulated radar data for 150 and 335 MHz (150 MHz for the EMR profile, and 335 MHz for the MCoRDS wideband profile) based on measured conductivity…"

Done.

Line 148 explain the acronym: "…Finite-Difference Time-Domain (FDTD) method…"

Done.

Line 165: Maybe say "We used a constant dielectric permittivity value of $\varepsilon'_r$ = 3.145 and EM wave propagation velocity of $\sim 1.69 \cdot 10^8$ m s$^{-1}$." to avoid repetition.

The problem is that these are not two different choices and the EM wave propagation velocity is just the consequence of the choice of permittivity.

Line 231-235: refer to a figure here?

Done.

**RC2**: 'Comment on egusphere-2024-2349', Marie G. P. Cavitte, 13 Oct 2024

**Review of "Age-depth distribution in western Dronning Maud Land, East Antarctica, from three decades of radar surveys" by Steven Franke et al.**

In this study, the authors trace nine IRHs over western Dronning Maud Land, with radar transects gathered over three different radar systems. They date them at the EDML ice core, using a combination of forward DEP modeling, as well as twtt-to-depth conversion using wave propagation and firn corrections. They then describe the IRH depths and geometries for the whole survey region and highlight the overlap between the IRHs traced here and other tracing studies in East and West Antarctica, showing very good promise for the AntArchitecture endeavour.

This paper is a very important contribution to making sure the interpreted internal stratigraphy of the Antarctic Ice Sheet is published and accessible to the community. Such efforts to document data sets should be praised and encouraged. However, I would suggest to consider submitting this article to ESSD, not for lack of quality of this manuscript, but because it seems like a better fit for a dataset paper. See minor comments below. I suggest this paper be published, here or in ESSD with technical revisions.

> We would like to thank Marie Cavitte for taking the time to review our paper and appreciate the constructive suggestions and the overall positive assessment. Below, we share our thoughts on the suggestion regarding whether the paper might be better suited for a data journal (e.g., ESSD).
>
> First of all, we fully understand the concerns and recognize valid reasons for submitting to either type of journal. A key aspect is whether the traced IRHs and their depth distribution in the ice sheet are viewed primarily as data or as a form of results. In our opinion, both perspectives can apply. Looking at similar manuscripts in the literature, we see that they have been published in both scientific journals and data journals. However, we acknowledge that our paper, in its current form, is mainly data-driven.
>
> Considering M. Cavitte's review, along with the other reviews and discussions with the editor into account, we have decided to expand the scientific focus to ensure the paper is suitable for *The Cryosphere* (TC). This expansion will primarily involve deepening the discussion of IRH depths (or normalized depths) in relation to glaciological aspects such as ice flow velocity, bed topography, and accumulation. Additionally, we will expand the discussion to include comparisons with similar IRHs from other studies, and we will introduce a new figure illustrating these connections.

**Specific comments:**

The first sentence of the introduction is very vague, and therefore not so useful, particularly "observing and modelling" which can encompass everything.

> We agree and the first sentence now reads as follows:
> *"Studying the dynamics of the Antarctic ice sheet (AIS) through a combination of geophysical observations and ice-sheet modeling is crucial for understanding its response to climate change and better projecting future sea-level rise."*

The impact of the firn correction of 13m taken as constant for the whole survey region should be discussed, as the snowfall regimes are quite different across it.

> Fully agreed and we added the following sentence in Section 2.5 (IRH depth and normalized depth):
> *"However, even though a 13 m firn correction was applied uniformly, snowfall and accumulation rates vary across western Dronning Maud Land (e.g., Rotschky et al., 2007) potentially affecting depth calculations of IRHs."*

> Furthermore, we have expanded our discussion on the topic of firn correction uncertainty in our paper.

**Line comments:**

Abstract – The last sentence uses "fundamental data", I would suggest "boundary conditions" instead.

> Done.

L16 – Comprehend → Understand

> Done.

L17 – Maybe specific where the melting is occurring (basal, surface)

> Done.

L21 – hundreds thousands → Hundreds of thousands

> Done.

L27 – linked to conductivity contrasts and density

> Done.

L28 – what does "detected across the ice sheet" mean? Reword

> Added "*thousands of kilometres*".

L30 – what is mean by boundary layers? And why use the word layer here? Not defined

> Good point. We rephrased the sentence to:
> *"Thus, each IRH represents an isochronous interface, originating at the ice-sheet paleosurfaces, where dielectric properties in the ice change."*

L30 – suggest to change "time horizons of the same of snow deposits" to "synchronous snow deposits"

> Done. We changed the sentence to:
> *"Moreover, IRHs are often considered isochronous because they form simultaneously at the ice-sheet surface and are then buried under subsequent snow accumulation, carrying with them changes in dielectric properties which radar systems are sensitive to (Bingham et al., 2024)."*

> We deviate from the wording "snow deposits" because as each IRH is the boundary between simultaneous snow deposits of different dielectric values and one can have dry deposition of impurities which are not related to snow deposits.

Table 1 – Developper → Developer

> Done.

L39 – why use the word "layer" and not "IRH" ?

> Done.

L44 – Suggest to modify to "where reflections have a different radar signatures due to their different vertical wavelengths"

> Thank you for the suggestion. Done.

L75 – the ice internal structure

> Done.

L84 – which serves as a transmit and receive

> Done.

L88 – Define fk

> Done.

L109 – what is meant by the final sentence? Clarify

> We want to highlight with this sentence that we show partially existing data (from Winter et al., 2019), e.g., their 38 and 74 ka IRHs and want to clarify with this sentence that we did not pick them ourselves but integrate these data into our study. We modified the sentence slightly and hope it is clearer now.

L175 – mention Ey after "electric field envelope"

> Done.

L235 – abscent → absent

Done.

L239 – number of data points does not represent much for the readers…I think it could be left out.

We deleted "points", however, on the other hand it is now unclear what the increase in data refers to.

L362- The review paper of AntArchitecture can now be cited here (https://egusphere.copernicus.org/preprints/2024/egusphere-2024-2593/)

Done.

Figure 1 – The Dome Fuji survey is missing. Also why aren't the EMR long-pulse lines drawn on this map, as drawn later on Fig.7. It's confusing to have different datasets on these two sets of figures. I would suggest to also mention in the figure caption what the background map is and also that ice core sites are highlighted with circles. Finally, the three shades of dark blue are really difficult to tell from each other on printed paper.

The purpose of Figure 1 is to show and focus on those radar lines that have been used in this study and their coverage. We explicitly want to avoid the expectation that tracing these nine IRHs is possible in all AWI lines. Hence, it covers only EMR short-pulse and UWB data and not the Dome Fuji EMR long-pulse.

We agree with the reviewer and follow the suggestion about mentioning the background map and also the ice core site markers in the figure caption. We also changed the color map on panel (b).

Figure 2 – the ice base reflection marker on panel c is not visible

We are not sure if the reviewer is referring to the marking symbol (the white arrow) or the bed reflection the arrow is pointing to. From our perspective both are visible in the radargrams (middle panel "335 MHz radar" in c). If the reviewer is referring to the modeled radar data, there is no bed reflection visible because it is not included in the model.

Figure 5 – Could Jutulstraumen be added to this figure too as it is discussed? It is helpful to have all the info on that one figure. Also, the figure caption could mention the different Features highlighted. Same for Figure 6.

Done.